# Benefits of the Neurogenic Potential of Melatonin for Treating Neurological and Neuropsychiatric Disorders

**DOI:** 10.3390/ijms24054803

**Published:** 2023-03-02

**Authors:** Yaiza Potes, Cristina Cachán-Vega, Eduardo Antuña, Claudia García-González, Nerea Menéndez-Coto, Jose Antonio Boga, José Gutiérrez-Rodríguez, Manuel Bermúdez, Verónica Sierra, Ignacio Vega-Naredo, Ana Coto-Montes, Beatriz Caballero

**Affiliations:** 1Department of Morphology and Cell Biology, Faculty of Medicine, University of Oviedo, 33006 Oviedo, Asturias, Spain; 2Instituto de Investigación Sanitaria del Principado de Asturias (ISPA), 33011 Oviedo, Asturias, Spain; 3Instituto de Neurociencias del Principado de Asturias (INEUROPA), 33006 Oviedo, Asturias, Spain; 4Servicio Regional de Investigación y Desarrollo Agroalimentario (SERIDA), 33300 Villaviciosa, Asturias, Spain

**Keywords:** melatonin, neural stem cells, adult hippocampal neurogenesis, neurological disorders, neurodegeneration

## Abstract

There are several neurological diseases under which processes related to adult brain neurogenesis, such cell proliferation, neural differentiation and neuronal maturation, are affected. Melatonin can exert a relevant benefit for treating neurological disorders, given its well-known antioxidant and anti-inflammatory properties as well as its pro-survival effects. In addition, melatonin is able to modulate cell proliferation and neural differentiation processes in neural stem/progenitor cells while improving neuronal maturation of neural precursor cells and newly created postmitotic neurons. Thus, melatonin shows relevant pro-neurogenic properties that may have benefits for neurological conditions associated with impairments in adult brain neurogenesis. For instance, the anti-aging properties of melatonin seem to be linked to its neurogenic properties. Modulation of neurogenesis by melatonin is beneficial under conditions of stress, anxiety and depression as well as for the ischemic brain or after a brain stroke. Pro-neurogenic actions of melatonin may also be beneficial for treating dementias, after a traumatic brain injury, and under conditions of epilepsy, schizophrenia and amyotrophic lateral sclerosis. Melatonin may represent a pro-neurogenic treatment effective for retarding the progression of neuropathology associated with Down syndrome. Finally, more studies are necessary to elucidate the benefits of melatonin treatments under brain disorders related to impairments in glucose and insulin homeostasis.

## 1. Melatonin

Melatonin (N-acetyl-5-methoxytryptamine) is a multifunctional hormone naturally produced and released rhythmically throughout the night by the pineal gland to regulate sleep–wake cycles [1]. The secretion of this neurohormone increases after the onset of darkness, reaching peak levels in the middle of the night, and gradually decreases in the second half of the night [2]. Light exposure stimulates the inhibition of melatonin production, and consequently, during the day its levels drop, becoming undetectable [3]. Furthermore, melatonin, once it is synthesized by the pineal gland, is promptly released into the bloodstream and is distributed among all tissues [4]. In particular, given the amphiphilic properties of melatonin, this neurohormone is able to cross biological barriers and enter cells, influencing tissue functions [5]. Additionally, melatonin is also locally synthetized in numerous cells and tissues, which presumably do not follow circadian rhythms.

Although melatonin is mainly referred to as the sleep hormone, this indolamine has been shown to exert antioxidant, anti-inflammatory and anti-apoptotic reprogramming in cellular homeostasis and disease (Figure 1). Melatonin mainly mediates its effects through MT1 and MT2 receptors, which belong to the superfamily of G protein-coupled receptors (GPCRs), by switching on/off intracellular signaling cascades. However, melatonin secretion, as well as the expression of melatonin receptors, has been widely proven to progressively decrease over the lifespan and in certain diseases [6,7], including neurodegenerative diseases or mental disorders [8]. This indicates that the downregulation of melatonin levels and their potential therapeutic effects may be involved in the onset and progression of diverse human diseases. Indeed, several recent studies endorse melatonin administration potentiality in various diseases, especially in neurodegenerative disorders [9,10].

## 2. Regulatory Role of Melatonin in Physiological Processes

### 2.1. Antioxidant Activity

Melatonin and its metabolic derivatives have been demonstrated to possess strong antioxidant properties against free radicals and are the reference agents in this field. This indolamine is considered an efficient scavenger of reactive oxygen species (ROS), reactive nitrogen species (RNS) and other oxidative agents [11]. Melatonin’s functions as an antioxidant include direct scavenging of free radicals, stimulation of the activity and efficiency of antioxidant enzymes, lowering the activation of pro-oxidant enzymes and improving the efficiency of mitochondrial respiration, thereby reducing ROS production [12]. First, melatonin, as an electron-rich molecule, acts as a potent endogenous free radical scavenger, forming stable end-products that are ultimately excreted by the organism. Additionally, this indolamine has been found to trigger the gene expression and activity of numerous antioxidant enzymes, such as superoxide dismutase (SOD), catalase (CAT), glutathione peroxidase (GPx) and glutathione reductase (GR), among others [13,14]; it therefore contributes indirectly to the detoxification of free radicals. Furthermore, melatonin also protects against enzymes involved in the generation of free radicals. Abundant evidence indicates that melatonin inhibits nitric oxide synthase (NOS) activity, xanthine oxidase (XO) and myeloperoxidase (MPO) [15,16,17]. Finally, it should be noted that melatonin also acts within the mitochondrion, an organelle that is widely considered the major intracellular source of ROS production. In this context, the mechanisms by which melatonin protects mitochondria involve multiple pathways, from the increase in the activity of antioxidant enzymes while reducing pro-oxidant enzymes within the mitochondria to the stabilization of the mitochondrial inner membrane, the improvement of the oxidative phosphorylation and the reduction of electron transport leakage ROS production and the control of opening the mitochondrial permeability transition pore [12]. Thus, taking all these factors into account, melatonin not only contributes both directly and indirectly to the detoxification of free radicals, but it also avoids their production, favoring the maintenance of cellular homeostasis.

### 2.2. Immune System Properties

One of the most relevant pleiotropic effects of melatonin is the modulation of the immune system, reducing chronic and acute inflammation [18,19]. This relationship is mainly established through the bidirectional communication between the pineal gland, as a neuroendocrine interface, and the immune system. As early as 1926, Berman described for the first time this potential interrelation, evidenced after kittens were fed pineal glands from bullocks, and observed increased learning, activity and resistance against infectious diseases. After this finding, numerous investigations focused on better understanding this relationship. In particular, to discover this tight connection, two experimental approaches have mainly been addressed: (a) pinealectomy and (b) the synchronization between rhythmic melatonin synthesis and the immune response [20,21]. Largely, pinealectomy causes a reduction in the size of both primary and secondary lymphoid organs, ultimately affecting innate responses [22]. Moreover, the circadian rhythm for the release of melatonin also influences antibody formation. When melatonin levels are higher, there is greater stimulation of antibody formation, thus contributing to an augmented immune response [23,24].

Generally, basic and clinical research suggests that the anti-inflammatory effects of melatonin are mediated by the modulation of anti- and proinflammatory cytokines [25,26]. Melatonin was reported to inhibit the production of two of the main inflammatory mediators, cyclooxygenase (COX) and inducible NOS (iNOS), by modulating nuclear factor kappa B (NF-κB) translocation [27,28,29,30]. In addition, this indolamine has also been shown to alleviate inflammasome activation. Melatonin was demonstrated to reduce lipopolysaccharide (LPS)-induced inflammation and the formation of the NLRP3 inflammasome in mouse adipocytes, thus inhibiting caspase-1 and IL-1 activation and the NLRP3 inflammasome-mediated pyroptosis [31,32]. Given that inflammation is implicated in the development and progression of several diseases, such as neurological disorders, the immunomodulatory effect of melatonin has gained increasing attention in recent years.

### 2.3. Anti-apoptotic Activity

Melatonin is considered a master regulator of cell death via the inhibition of apoptotic responses and the activation of survival pathways. Mitochondria are one of the main cellular organelles that sense and respond to many stressors, leading to adaptive and maladaptive responses through the regulation of diverse signaling pathways, among which it is worth highlighting apoptosis and autophagy [33]. Under adverse conditions, mitochondrial function is affected, which ultimately triggers the release of cytochrome c (CytC) and apoptosis-inducing factor (AIF) into the cytosol, and therefore, the activation of apoptosis machinery. Caspases are also critical regulatory molecules that generate a cascade of signaling events, controlling cell death in disease. It has been found that melatonin is capable repressing the mitochondrial-mediated apoptotic response by enhancing a compensatory pathway. This neurohormone increases the expression of the anti-apoptotic Bcl-2 family proteins, but it inhibits the activity of the pro-apoptotic Bax protein by acting on the SIRT1/NF-κB axis [34,35]. Additionally, melatonin also suppresses caspase-dependent apoptosis [36]. This indolamine was found to silence the caspase-1 pathway and reduce the overexpression and activation of caspase-3 [37,38]. Furthermore, melatonin not only suppresses apoptotic responses but also stimulates the Akt pathway, mediating the activation of autophagy and its roles in cell survival [10,39]. Therefore, melatonin has been widely demonstrated to be a potent anti-apoptotic agent mainly due to its regulatory action on proteins involved in mitochondria-mediated apoptosis and on cell survival mechanisms.

## 3. Neurogenesis in the Adult Brain

The current dogma “that new neurons can and do form in the adult mammalian brain” is well accepted [40]. In the adult brain, neurogenesis is a process that starts with cell proliferation and ends with new functional neurons that integrate into existing neural circuits. There are two “canonical” regions of the mammalian adult brain that generate new neurons: (a) the border of the lateral ventricles of the brain (subventricular zone) and (b) the subgranular zone of the hippocampal dentate gyrus [41,42,43,44]. Several non-canonical regions also contain neural progenitor cells, including the neocortex, striatum and hypothalamus [44].

In adult hippocampal neurogenesis, the differentiation of adult neural stem cells into mature functional neurons proceeds through a clearly defined set of cellular stages starting from (a) type-1 cells (radial glia-like cells) that express different markers such as glial fibrillar acidic protein (GFAP), nestin, SRY-box transcription factor 2 (SOX2) and brain lipid-binding protein (BLBP) to (b) type-2a and 2b cells (transiently amplifying progenitor cells) that are positive for specific markers (e.g., nestin, SOX2, achaete-scute family bHLH transcription factor 1 (ASCL1), T-box transcriptional factor 2 (TBR-2), etc.); and (c) type-3 cells (neuroblasts), which undergo migration and final maturation to functional neurons and express specific markers such as polysialylated neural cell adhesion molecule (PSA-NCAM) and neurogenic differentiation factor 1 (NeuroD1) [41,42,44]. The newly created neurons developing from neural stem cells in the hippocampus integrate into pre-existing neural networks of the granular neuron layer of the dentate gyrus to participate in learning and memory processes [43,44,45].

In addition to ciliated ependymal cells, the subventricular zone of lateral ventricles presents a varied niche of neural stem cells and precursors, including (a) proliferating neuroblasts (type A cells) that express different neuronal markers (Tubulin beta 3 Class III (TUBB3) and NeuroD1) and are able to migrate to the olfactory bulb via the rostral migratory pathway (b) slowly proliferating cells (type B cells), which express multipotent neural stem cell markers (nestin and GFAP) and show capacity for self-renewal and differentiation toward neurons and glial cells; and (c) transiently amplifying progenitors (type C cells) that show nestin expression and a very active state of proliferation [41].

Neurogenesis alteration can be a consequence of a decrease in the pool of neural stem cells, alterations in the molecular microenvironment that do not favor cell proliferation and/or cell differentiation or because neural stem cells and progenitor cells cannot respond to neurogenic signals in the aged brain and/or under neurodegeneration [42,46]. Therefore, interventions that can maintain adult brain neurogenesis are key to improving neurological functions, even in the late phases of aging, and especially under conditions of neurodegeneration [42,43,45,46].

Melatonin promotes neuroprotection due to its antioxidant, anti-inflammatory and anti-apoptotic properties [10,47,48,49,50]. Melatonin also plays an important role in the regulation of neurogenesis [51,52,53]. Therefore, melatonin is a potential treatment for neurodegenerative and neurologic disorders associated with an impairment of neurogenesis, including normal brain aging, dementia, stress, depression, stroke, traumatic brain injury, etc. [53,54,55]. The first evidence of the neurogenic potential of melatonin comes from studies in pinealectomized rats, which showed that an important reduction in melatonin levels also leads to decreases in adult hippocampal neurogenesis [56]. Exogenous administration of melatonin in these animals restored their melatonin levels and the functionality of neurogenesis [56]. Similarly, chronic administration of luzindole (an antagonist for melatonin receptors) also demonstrated that a lack of melatonin significantly affects hippocampal neurogenesis in C57BL/6 adult mice [57].

Melatonin is able to modulate several processes involved in adult brain neurogenesis including survival, proliferation and neuronal differentiation processes of neural stem/progenitor cells, normal migration of neuronal precursors and survival and maturation of newly created neurons (formation and growth of dendrites, complexity of dendrite trees, length of axonal prolongations, processes of synaptic plasticity, etc.) [42,46,51,55,58,59,60,61,62]. These neurogenic actions of melatonin may involve several signaling pathways and molecular mechanisms that impact adult brain neurogenesis [53,55,62,63], as summarized in Figure 2.

Neurogenic pathways of melatonin can be mediated by its membrane receptors (MT1/2) [51,57]. Signaling pathways related to activation of mitogen-activated protein kinases (MAPK) and extracellular signal-regulated kinases 1/2 (ERK 1/2) as well as phosphoinositide 3-kinase (PI3K) and protein kinase B (Akt) are frequently involved in melatonin-receptor-related effects on cell survival, proliferation and neuronal differentiation of neural stem/progenitor cells [51,52,53]. However, other neurogenic actions of melatonin can be independent of its receptors, including melatonin’s capacity to promote cell survival of neural progenitor and precursor cells through regulation of apoptosis by modulating Bcl-2 family proteins [51,53], and due to its antioxidant properties by modulation of Nrf2 signaling [51,53]. Increases in neurotrophic factors, such as brain-derived neurotrophic factor (BDNG), are also involved in the capacity of melatonin to activate neuronal differentiation processes in neural stem/progenitor cells independently of its receptors [51,53]. Histone acetylation is also promoted by melatonin to activate neuronal differentiation, which may or may not involve its membrane receptors [51]. Melatonin can activate tropomycin-receptor kinase B (TrkB) signaling via the MT1 receptor or by increasing BDNF levels to impact cell survival and proliferation [51,52,53]. Finally, melatonin also promotes maturation processes in new postmitotic neurons, including the formation, growth and complexity of dendrites and synaptic plasticity [59,61]. In this last case, melatonin is able to directly interact with intracellular proteins, such as calcium/calmodulin-dependent kinase II (Ca^2+/^CaMKII) and the Ca^2+^ binding protein calretinin, to modulate the cytoskeleton and promote different neuronal maturation processes [64,65,66] (Figure 2).

These beneficial effects of melatonin on different parameters of neurogenesis (survival, proliferation, differentiation, maturation, etc.) have been demonstrated by using different concentrations of melatonin (from nM to μM) during acute treatments (from hours to days) in several models in vitro, including commercial cell lines (e.g., rat PC12 and mouse C17.2 cell lines), primary cultures of neural stem/progenitor cells obtained from the brain or spinal cord of adult rats and mice, cultured mesenchymal stem cells from human amniotic fluid, induced pluripotent stem cells from mice and adult rat/mouse hippocampus organotypic cultures [55]. Likewise, several animal models have confirmed the neurogenic potential of melatonin in both acute and chronic melatonin treatments [55]. For instance, various studies in Balb/C mice have demonstrated positive effects of melatonin promoting the proliferation and survival of neural progenitor cells as well as the survival, maturation and complexity of dendrites in the new postmitotic immature neurons in the dentate gyrus of the hippocampus of these adult mice [54,67,68]. Melatonin also modulates the structural plasticity of axons in granule cells in the dentate gyrus of Balb/C mice by regulating mossy fiber projections to establish new functional synapses in the hippocampus [61]. Similarly, melatonin also increased the survival of neural progenitor cells and postmitotic immature neurons in the dentate gyrus of adult C57BL/6 mice [58]. More recent studies have shown that melatonin restores the functionality of adult hippocampal neurogenesis during the accelerated and pathological brain aging of SAMP8 mice [46]. The benefits of melatonin acquire an especial relevance in the success of treatments of different nervous tissue lesions by transplants with mesenchymal stem cells. Antioxidant and anti-inflammatory properties of melatonin may improve the survival and functionality of transplanted mesenchymal stem cells. This then leads to favorable outcomes in different experimental treatments, for instance, in focal cerebral ischemia and neurodegenerative processes of Alzheimer’s disease [62]. Neurogenic differentiation from mesenchymal stem cells specifically requires wingless-integration-1 (Wnt) expression and activation of c-Jun N-terminal kinases (JNK) pathway [62]. Finally, melatonin is also able to revert those alterations on neurogenesis that are induced by several drugs or toxic compounds including lipopolysaccharides, valproic acid, methamphetamines, cuprizone, methotrexate, dexamethasone, metformin, scopolamine, 5-fluorouracil and corticosteroids [53,55,62,69,70,71,72,73,74].

## 4. Melatonin and Neurogenesis: Impact on Different Neurological Disorders

### 4.1. Aging and Dementia

Neurogenic processes can persist in the adult aged brain [75]. However, these are altered during the normal aging of the brain as well as under different neuropathological conditions, such as dementia [42,43,45,55,76]. Various studies in animal models of aging have demonstrated the beneficial effects of melatonin associated with an improvement in neurogenesis. For instance, acute treatments with melatonin improved neurogenesis in C57BL/6 mice with d-galactose-induced aging, since melatonin restored proliferation and neuronal differentiation processes while improving spatial memory deficits in these aged mice [77]. Likewise, long term treatments with melatonin for 3, 6 or 9 months were able to positively modulate hippocampal neurogenic processes during the normal aging of Balb/C mice [54,64]. In this study, melatonin increased cell proliferation (>90%), promoted the survival of newly created cells (>50%) and increased the number of doublecortin-labeled postmitotic cells (>150%) and calretinin-positive neurons (>66%) in the dentate gyrus of the hippocampus of these mice [54,64]. Therefore, the anti-aging properties of melatonin seem to be linked to its neurogenic properties.

Alzheimer´s disease (AD) is the most common form of dementia during aging. Importantly, the age-related decline in adult hippocampal neurogenesis can be accelerated due to the presence of AD, thus further contributing to hippocampus-dependent cognitive and emotional dysfunctions during aging [45,75]. Likewise, recent studies have shown a compromised adult hippocampal neurogenesis in the aged and neurodegenerative brains of SAMP8 mice [46]. These mice are considered an animal model for studying AD-related neurodegenerative processes [47,49,78]. Chronic treatment with melatonin (10 mg/kg) for 9 months in SAMP8 mice improved cell survival, the correct migration of neural precursor cells and their neuronal maturation (e.g., neural processes and the length of neuronal prolongations) [46]. Given that SAMP8 mice present an early decrease in melatonin levels and its receptors in several organs, including the brain [47,79], the benefits and neurogenic actions of melatonin in these mice seem to be independent of the melatonin receptors.

### 4.2. Stress, Anxiety and Depression

The most common mental disorders are mood disorders, especially depression and situations of chronic stress and anxiety. In particular, depression is one of the most frequent mood disorders in the world [80,81]. A depressive mood may be an adaptive stress response manifested by different behavioral phenomena, such as anhedonia, psychomotor disturbance and loss of appetite and sleep, that can lead to a great variety of psychological symptoms ranging from loss of motivation and energy to suicidal thoughts [81,82]. Theories involving the presence of neuroinflammation and impairments in neurogenesis and processes of neuronal remodeling are widely accepted in the pathogenesis related to mood depression [81].

The antidepressant-like and anti-stress effects of melatonin are well known [58,67,68,82,83,84]. Accordingly, melatonin levels decrease in the plasma of depressive patients [83]. Several studies regarding stress, anxiety and/or depression have confirmed the antidepressant properties of melatonin linked to its neurogenic potential [82]. For instance, treatment with melatonin (2.5 mg/kg) in BalB/C mice exposed to chronic mild stress showed relevant antidepressant-like effects by inducing relevant anti-inflammatory and neurogenic responses in the dentate gyrus of the hippocampus. The neurogenic benefits of melatonin in these mice included increased cell proliferation and survival as well as a higher complexity of the dendrite trees in doublecortin-positive neurons [67,68]. Ramirez-Rodriguez et al. (2020) also confirmed the interrelationship between antidepressive and pro-neurogenic properties of melatonin by applying different melatonin concentrations (ranging from 0.5 to 10 mg/kg) to Balb/C mice for 14 days. These authors observed that melatonin (>2.5 mg/kg) was able to significantly increase the number of doublecortin-positive neurons as well as the number and complexity of dendrite trees in the hippocampal dentate gyrus while decreasing the immobility behavior of mice in forced swim tests [67]. Similar effects were found in C57BL/6 mice treated with melatonin (8 mg/kg) since this indolamine promoted the survival of neural progenitor cells and newly postmitotic immature neurons while also decreasing depressive-like behavior in the forced swim test [58]. Other animal models based on maternal separation of infants and their social isolation are also used as study models of depression. These animal models showed disruptions of cell proliferation and the creation of new neurons in the hippocampal dentate gyrus, which were reverted by exogenous administration of melatonin (10 mg/kg) [85]. Melatonin also potentiates the beneficial effects of other common antidepressant drugs, such as citalopram and ketamine, by significantly modulating the adult hippocampal neurogenesis (proliferation, survival, number of newly created neurons, etc.), while minimizing their possible secondary effects (e.g., increase in locomotor activity) [80,83]. Agomelatine, a common antidepressant drug that functions via melatonin receptors, also exerts beneficial effects on adult hippocampal neurogenesis. This drug promotes cell survival and proliferation in the dentate gyrus of stressed rats and with anxiety due to their light exposure twice a day for one week [86]. Agomelatine also favors neuronal maturation of dendrite trees in newly created doublecortin-positive neurons in stressed rats and with a depressive mood due to chronic and constant exposure to light [87]. Therefore, modulation of neurogenesis by melatonin is clearly beneficial under conditions of stress, anxiety and depression.

### 4.3. Acquired Brain Damage

Different brain damage can be acquired during adult life, such as traumatic brain injury (TBI), damage under situations of ischemia and reperfusion as well as consequences of brain stroke. Brain injuries can activate a neurogenic response in the brain by promoting the proliferation of neural stem cells and their migration to the injured area to initiate recovery of the affected nervous tissue [88]. In the particular case of TBI, the neurogenic response is limited, depending on the level of brain damage acquired and patient age, and in the best cases, this response is not extended more than 14 days [89,90]. In this way, therapeutic strategies based on melatonin administration [91,92,93] and/or stem cells are promising approaches in several animal models of TBI [89,90,94]. For instance, melatonin attenuates deficits in spatial memory and motor function in mice with cortical compact injury via the modification of cortical and hippocampal dendritic spine morphology, decreasing hippocampal microgliosis and neuroinflammation, and promoting neurogenesis [94]. In a recent study, a three-dimensional system containing the combination of neural stem cells and melatonin repaired damage from TBI in rats so that the brain injury volume decreased, while cell survival increased with an important recovery of neurologic functions [88]. This last study also shows that melatonin is able to promote in vitro the differentiation of neural stem cells in that matrix into different cell types of the nervous system [88].

Melatonin also protects against brain ischemia/reperfusion injuries and stroke [95,96,97,98]. The beneficial effects of melatonin under ischemic and brain stroke injuries include antioxidant and anti-inflammatory responses, preservation of the blood–brain barrier and an improvement in neurobehavioral outcomes [95,96,97]. Neurogenesis is also increased by melatonin in ischemic-stroke mice via its melatonin receptors. Under this condition, melatonin promotes the proliferation of neural stem cells and their neuronal differentiation, thus activating structural and functional recovery of the ischemic and/or infarcted brain area [96,97,98]. Likewise, melatonin pretreatment also improves the survival and function of transplanted mesenchymal stem cells after focal cerebral ischemia. In particular, melatonin promotes neurogenesis and angiogenesis processes in transplanted mesenchymal stem cells, resulting in a decrease in the brain infracted area and an improvement in neurobehavioral outcomes [99]. Given these premises, the pro-neurogenic properties of melatonin have relevant benefits for the ischemic brain or after a brain stroke.

### 4.4. Down Syndrome

Down syndrome (DS) is a consequence of a trisomy in chromosome 21 and is the most common cause of mental retardation by chromosome disorder. Ts65Dn mice are a good animal model of DS with developmental delays and motor, cognitive and behavioral alterations similar to those of patients with DS [100,101,102]. Specifically, cognitive impairments in these animals include deficits in hippocampal-dependent learning and memory processes as a consequence of impaired neurogenesis, relevant hypocellularity, increased oxidative stress and several other neuromorphological and electrophysiological alterations [101,102].

Long term treatment with melatonin has been associated with relevant antioxidant and anti-aging effects in Ts65Dn adult mice [103]. Likewise, chronic oral supplementation with melatonin is also able to improve spatial learning and memory in middle-aged Ts65Dn mice, thus delaying the neurodegeneration and cognitive deterioration that characterize these animals [100]. These beneficial effects of melatonin in Ts65Dn mice are related to an improvement in adult neurogenesis, since melatonin increases cell proliferation and promotes the differentiation of neuroblasts in the hippocampus of these mice [101]. However, melatonin has neither pro-neurogenic effects nor prevents the cognitive impairments of Ts65Dn mice when it is administered in pregnant animals throughout their conception as well as in their offspring until 5 postnatal months [102]. Therefore, melatonin may represent an effective pro-neurogenic treatment, at least for retarding the progression of neuropathology of DS in the adult brain but not during neurodevelopment.

### 4.5. Epilepsy

Epilepsy is a neurological disorder characterized by anomalous brain activity that arises because of a burst of abnormal electrical activity, ultimately causing seizures and sometimes loss of awareness. Epilepsy is the second most common chronic neurological disorder after headaches and affects all age groups. Globally, approximately 5 million people are diagnosed with epilepsy each year, and currently, over 70 million people suffer from this disorder [104,105]. Generally, epilepsy is treated by taking a combination of drugs, commonly known as antiepileptic medication, which can reduce the frequency and intensity of seizures. Currently, more than 20 different types of antiepileptic medications are available, but these treatments are not always effective, and sometimes surgery is recommended [106,107]. Although important research advances have been made in the development of more effective drugs for the treatment of epilepsy during the last few years, the vast majority of medicaments used today cause apoptotic neurodegeneration [108], and approximately 35% of patients experience recurrent spontaneous seizures [109].

Increasing lines of evidence indicate that melatonin can act as an adjunctive treatment in epilepsy given its neuroprotective effects [110]. Generally, epileptic disorders were found to induce excitotoxic neural death mediated by the interaction of the GluR2/GAPDH complex. A study carried out in a rat model revealed that chronic treatment with melatonin after induction of epilepticus reduces seizure severity and neural death by disrupting the GluR2/GAPDH complex interaction [111]. Likewise, it has also been described that this neurohormone mediates an anticonvulsant effect by acting on GABA receptors [112]. Melatonin was found to block voltage-dependent calcium-channel-mediated neurotransmitter release, leading to the suppression of neural activity and, therefore, reducing epileptic symptoms [113]. It is worth mentioning that the use of melatonin as a possible treatment for epilepsy is gaining even more attention given the regulatory properties of this indolamine in cell neurogenesis and differentiation. Uyanikgil and colleagues studied for the first time the modulatory action of melatonin administration on neurogenesis in newborns of pinealectomized rats subjected to epilepsy during pregnancy [114]. In the cerebellum of these animals, it was observed that the lack of melatonin leads to increased expression of nestin, a widely known marker of neural stem cells, which has been recently demonstrated to negatively regulate neural differentiation and survival [115]. Interestingly, melatonin administration was able to inhibit the disproportionate expression of nestin and alleviate epileptic seizures [114]. Later, another study also observed a similar response in the CA1 region of the hippocampus [116], further strengthening that melatonin exhibits neurogenic and neuroprotective properties in epilepsy.

Moreover, melatonin coadministration in rats receiving valproic acid (VPA) treatment was found to regulate certain adverse effects caused by the conventional antiepileptic drug used [117]. VPA was described to trigger oxidative stress, subsequently stimulating the expression of p21 and the induction of cell cycle arrest [118], which resulted in the repression of hippocampal neurogenesis [117]. However, melatonin coadministration protected against VPA-induced alteration of neurogenesis by downregulating p21 levels [117]. In line with these findings, melatonin appears to regulate neural migration and differentiation in epileptic disorders. It was shown that VPA treatment leads to cognitive impairments and a reduction in cell proliferation in the hippocampus by decreasing doublecortin (DCX) levels [119]. Melatonin administration was able to increase the protein levels of DCX, suggesting that this indolamine seems to regulate neurogenesis by influencing the proportion of newly generated immature neurons and, consequently, counteracting the cognitive impairment induced by VPA [117]. In addition, VPA was also found to decrease the expression of SRY-Box transcription factor 2 (SOX2) [120], which is tightly involved in the promotion of neural stem cell proliferation and survival and in the activation of neural differentiation [121,122]. Melatonin treatment had previously been shown to increase SOX2 levels, promoting cell proliferation [123]. Moreover, the adjuvant administration of melatonin in epilepsy was demonstrated to also upregulate SOX2 expression in the hippocampus, counteracting neurogenesis impairment [117]. Therefore, all these works indicate that melatonin positively regulates neuronal differentiation and survival in epilepsy through its action on different cell signaling pathways, and its use as an adjuvant to suppress the adverse aspects of the currently used antiepileptic drugs is interesting.

### 4.6. Schizophrenia

Schizophrenia is a severe mental disorder characterized by positive or psychotic symptoms, such as hallucinations, which are usually temporary, and by negative symptoms, such as anhedonia and cognitive dysfunction, that tend to be chronic [124,125,126]. Cognitive and social impairments are the first symptoms that patients experience, in a period called prodromal, many years (>10 years) before suffering the first episode of psychosis, which generally appears in adolescence or in early adulthood [125]. Although schizophrenia is not a common disease (incidence is approximately 1% of the world´s population [127]), acute schizophrenia is valued as one of the most disabling disorders worldwide [128], which makes its diagnosis crucial for early intervention. However, schizophrenia is currently diagnosed once the patient experiences two or more episodes of psychosis following the criteria described in the Statistical Manual (DSM) of the American Psychiatric Association [129] or in the International Classification of Diseases (ICD) of the World Health Organization [130]. Moreover, most of the currently existing antipsychotic drugs, which primarily mediate the blockage of neurotransmitter receptors [131], seem to be effective against positive symptoms, but exert only a slight improvement in negative and cognitive symptoms [132].

Currently, schizophrenia pathophysiology is unknown, but the evidence suggests that this disorder is caused by early brain developmental impairment due to the influence of genetic and environmental risk factors [126]. Interestingly, season of birth is associated with an increased risk of developing schizophrenia [133]. It has been suggested that the maternal chronobiological disturbances during pregnancy might be the cause [134]. Alterations in melatonin production during pregnancy were also related to psychiatric disorders [135]. Nevertheless, existing research on the effect of melatonin on neurodevelopment, particularly in schizophrenia, is limited [136]. A study carried out with human olfactory neuronal precursors (ONPs), a model widely used for the study of neurodevelopment since it shares many characteristics with embryonic neural stem cells [137], has shown that melatonin promotes neural differentiation [136]. Specifically, the differentiation capacity of ONPs derived from patients with schizophrenia was markedly reduced, which was found to be associated with decreased levels of the phosphorylated protein glycogen synthase 3β (GSK3β) and melatonergic receptors, which seem to be essential for axon formation [136]. Interestingly, melatonin was found to counteract this reduction in ONPs obtained from schizophrenic patients. Furthermore, these ONPs from patients diagnosed with schizophrenia also exhibited an alteration in the cellular mechanism of exocytosis, which was found to be improved with melatonin treatment, since this neurohormone causes changes in the actin microfilaments of the cytoskeleton [138]. These results suggest that melatonin is involved in neurodevelopment by promoting differentiation and the correct establishment of neuronal circuits. In the context of fetal brain development, the proper melatonin supply from the mother to the fetus is critical for the appropriate development of neuronal morphology and the functional differentiation of neurons, as well as for the prevention of schizophrenia and the associated neurodevelopmental anomalies [136,138]. Despite the recent findings, there remains a significant need for research in this direction to deepen the understanding of the origin of this pathology.

On the other hand, neurogenesis seems to be impaired during neurodevelopment in schizophrenia, and there is also a reduction in this capacity in schizophrenic adults [139], especially affecting the hippocampal stem cells of the dentate gyrus [140]. Additionally, structural and functional abnormalities of the hippocampus were found to be present in patients with schizophrenia, where hippocampal shrinkage, synaptic alterations and a disconnection with the rest of the central nervous system were observed, leading to the cognitive impairments experienced by these patients [141]. Several studies have shown that melatonin is able to attenuate alterations in the hippocampal region of the brain and improve the cognitive system [46,142,143] by promoting both proliferation and differentiation of neurons [142,143]. Moreover, patients with schizophrenia were found to exhibit a reduction in the size of the pineal gland, leading to disturbances in melatonin secretion levels [144]. However, many of the antipsychotics used for the treatment of this mental disorder, such as haloperidol and risperidone, do not improve the levels of melatonin or the levels of GAP-34, a protein involved in neurodevelopment and neuroplasticity [145]. Therefore, the administration of melatonin together with antipsychotics could improve the symptomatology of these individuals. Indeed, a recently published review has shown that the administration of melatonin together with antipsychotics improves sleep and medication side effects [146]. Moreover, although melatonin administration did not show significant improvement in cognitive function compared to control individuals, this neurohormone was able to improve the memory of patients with schizophrenia in comparison to baseline assessments [146]. Given that adjunctive melatonin therapy seems to exert positive outcomes in schizophrenia, future investigations using large sample sizes and testing melatonin administration together with different antipsychotic drugs are needed.

### 4.7. Amyotrophic Lateral Sclerosis (ALS)

ALS is a rare disease that causes the progressive deterioration and loss of function of motor neurons in the anterior horn of the spinal cord, eventually leading to severe disability and death. Normally, patients die from respiratory failure within 2–4 years after symptom onset [147]. Overall, the worldwide incidence of ALS is approximately 1.7 cases per 100,000, with a prevalence of 5 in 100,000 people [148]. Generally, the diagnosis of ALS, especially during the early stages of the disease, is complicated and uncertain. The major problem lies with the absence of characteristic abnormalities of the first and the second motor neuron pathways. As a result, there is no definitive test for ALS, and the diagnosis is determined by the exclusion of other causes of motor neuron dysfunction [148,149]. Moreover, since it is an extremely heterogeneous disease, the only approved drug to date is riluzole, a benzothiazole derivative that blocks glutamatergic neurotransmission in the central nervous system. Even though this treatment is well tolerated, its efficacy in ALS is moderate, prolonging the survival of patients by only 2–3 months [150,151].

Despite intensive research efforts, the etiology of ALS is currently not well understood. Research has indicated that the major pathophysiological mechanism linked to ALS is mutation in the SOD1 gene. Data obtained in a recent study support that folding intermediates, instead of the mature SOD1 protein, promotes the accumulation of toxic substances [152], which seem to be influenced by the decrease in the antioxidant cellular capacity by affecting zinc binding capacity [153,154]. Moreover, ALS also courses with protein aggregation, which involves cytoskeletal proteins, as well as mitochondrial dysfunction. These alterations, in turn, promote greater cytotoxicity [155].

In recent years, studies focused on evaluating the therapeutic action of melatonin in ALS have gained attention. Indeed, melatonin has been proposed as a potential compound for neuroprotection in ALS since it leads to better prognosis by slowing disease progression and prolonging survival [156]. Although at present there are no significant research reports on the effects of this indolamine on neural stem cells from subjects with ALS, various modulatory effects of melatonin on diverse cellular mechanisms that could influence cell fate and differentiation have been observed. The antioxidant properties of melatonin were found to allow the reduction of oxidative stress in ALS patients [157]. It has been widely described that the regulation of redox signaling is critical for the coordination of the cell cycle with differentiation to ensure homeostasis and control cell fate in other diseases and in developmental processes [158,159,160]. ROS directly regulate cellular signaling, especially metabolism and cell death processes. A novel study has recently shown that melatonin also promotes autophagy in ALS mice via the upregulation of SIRT1 [161]. Moreover, another work found that melatonin exerts neuroprotective effects in the mutant SOD1 (G93A) transgenic mouse model of ALS through the regulation of caspase-mediated cell death. Melatonin inhibited activation of the caspase-1 pathway, blocked the release of mitochondrial CytC and decreased the expression and activation of caspase-3 [37]. Several works have focused on evaluating the role of autophagy and cell death signaling in the modulation of neural stem cells. Autophagy was demonstrated to regulate Wnt and Notch signaling, two cellular pathways that are essential for adequate neural differentiation [162,163]. The available data denote that autophagy contributes to the preservation and activation of quiescent adult neural stem cells and the survival of newborn neurons [164]. Likewise, cell death pathways were found to act as regulatory mechanisms in neurogenesis and synaptic activity. The blockade of caspase-1 was reported to increase neurogenesis [165], and the local and controlled activation of caspase-3 mediated cell death was intimately involved in many regulatory mechanisms, including the regulation of neuron death during neurodevelopment and synapse pruning during differentiation [166]. Therefore, data obtained in relation to the regulatory action of melatonin on these mechanisms seem to suggest that melatonin may enhance neurogenesis in ALS not only by preventing neurotoxicity but also by acting as an autophagy activator and a cell death regulator. Thus, a better understanding of how melatonin acts on these mechanisms in ALS may provide insights into how melatonin could be used to reprogram neural progenitor cells for regeneration.

### 4.8. Other Common Diseases Related to Neurogenesis Impairments

Melatonin also attenuates neurogenesis impairments in a wide range of other common diseases. Among them, we would like to note diabetes mellitus (DM) given its high prevalence in the population worldwide. Importantly, this metabolic dysfunction may negatively impact the adult brain [167] and even contribute to the development of AD [168]. Exogenous administration of melatonin may alleviate DM and its related complications (e.g., diabetic neuropathies) in cell lines, animal models and diabetic patients [169]. For instance, leptin-deficient (ob/ob) mice are a preclinical model of obesity and type 2 diabetes, with clinical complications such as diabetic neuropathy [170,171]. In this way, these mice express markers of neurodegeneration and anxiety-like and stress-like behaviors [167]. Exogenous administration of melatonin (500 μg/kg) has beneficial effects on ob/ob mice, such that melatonin is able to modulate several parameters, including oxidative stress, stress of the endoplasmic reticulum, inflammation, adipogenesis and proteolytic systems in organs such as the liver and brain. As a consequence, melatonin reduces neurodegeneration in the brains of ob/ob mice and the behavioral impairments found in these animals [167,171,172]. In contrast, endogenous melatonin may interfere with measurements of glycemic control in diabetic patients [173]. Likewise, a rare variant of melatonin receptor 1b (MTNR1B) has been recently identified and is associated with impaired glucose tolerance and an increased risk of type 2 DM [169].

There are few studies about the pro-neurogenic effects of melatonin in situations of DM. However, it is well known that disruptions in the plasma levels of insulin and/or glucose may negatively impact not only neurodevelopment [174,175] but also adult brain neurogenesis in animal models of DM [176,177,178]. Oxidative stress and its consequences mediate the inhibitory effects of high levels glucose on the differentiation of neural stem cells [179]. Wongchitrat et al. (2016) observed that melatonin treatment (10 mg/kg) for 4 weeks in mice with DM decreased astrogliosis in the hippocampus and reverted impairments observed in neurogenesis and synaptogenesis via melatonin and insulin receptors, with an improvement in spatial memory deficits [55,98]. Other studies have also demonstrated beneficial effects of melatonin in animal models of gestational diabetes. Injections of melatonin (10 mg/kg) in these animals prevented fetal neuropathies in their offspring given that melatonin promoted the proliferation and survival of neural stem cells and avoided their premature neuronal differentiation [55,174,175]. Therefore, although the benefits of melatonin for DM are still controversial, the interplay between melatonin and neurogenesis is promising as a potential treatment for brain disorders related to impairments in glucose and insulin homeostasis. However, further studies are necessary to support this point.

## 5. Conclusions

Neuroprotective effects of melatonin on different neurological pathologies may include (a) improvements in antioxidant defenses; (b) a relevant anti-inflammatory response; (c) decreased cell sensitization to apoptosis; and (d) promotion of appropriate functional neurogenesis. These properties of melatonin may promote several benefits on neural stem cells, without toxic side effects, that include increases in the survival and proliferation of neural stem/progenitor cells as well as the promotion of neuronal differentiation processes of neural progenitor cells and the correct migration and neuronal maturation of neural precursor cells. Finally, melatonin treatments may improve neurobehavioral outcomes as well as motor and cognitive functions (e.g., learning and memory processes). Depending on the specific neurologic pathology, melatonin may revert and/or, at least, delay those alterations that affect neurogenic mechanisms in order to promote appropriate neurogenesis and the structural and functional recovery of those damages that are produced on the nervous tissue under different neuropathologies, as reviewed above. In particular, pro-neurogenic actions of melatonin may be beneficial during brain aging, dementias, situations of anxiety and chronic stress, under a depressive mood, after TBI, neurological damage due to ischemia and stroke, in Down syndrome, under conditions of epilepsy, schizophrenia and ALS and due to metabolic disorders with a relevant neurological impact, such as DM (Figure 3). In this way, future strategies based on appropriate melatonin administration might be useful in clinical practice to fight these common and frequent neuropathological conditions. It is paramount to note that we have always included both neurogenic niches of the adult brain in our searches regarding effects of melatonin on neurogenesis under different neurological and neuropsychiatric conditions. However, it is true that in most of the pathologies considered in our study, the neurogenic zone affected by melatonin has always been the stem cell niche of the hippocampus. Therefore, neural stem cells of the dentate gyrus of the hippocampus seem to be a key target for the benefits of melatonin in neurological diseases related to impairments of adult brain neurogenesis.

## Figures and Tables

**Figure 1 ijms-24-04803-f001:**
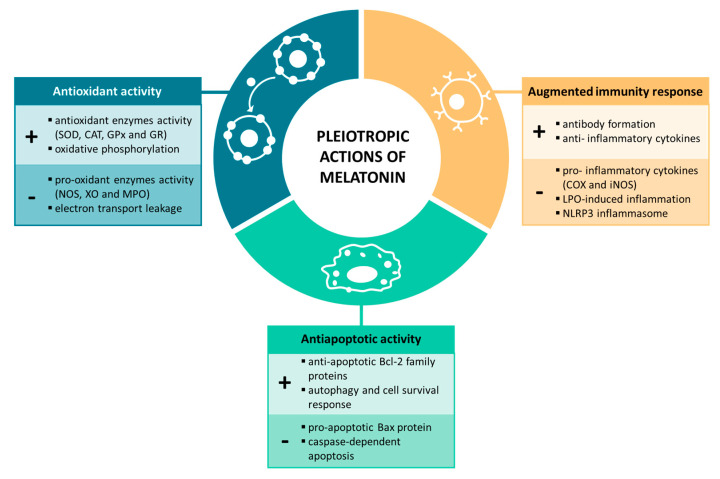
Melatonin’s pleiotropic actions on physiological processes.

**Figure 2 ijms-24-04803-f002:**
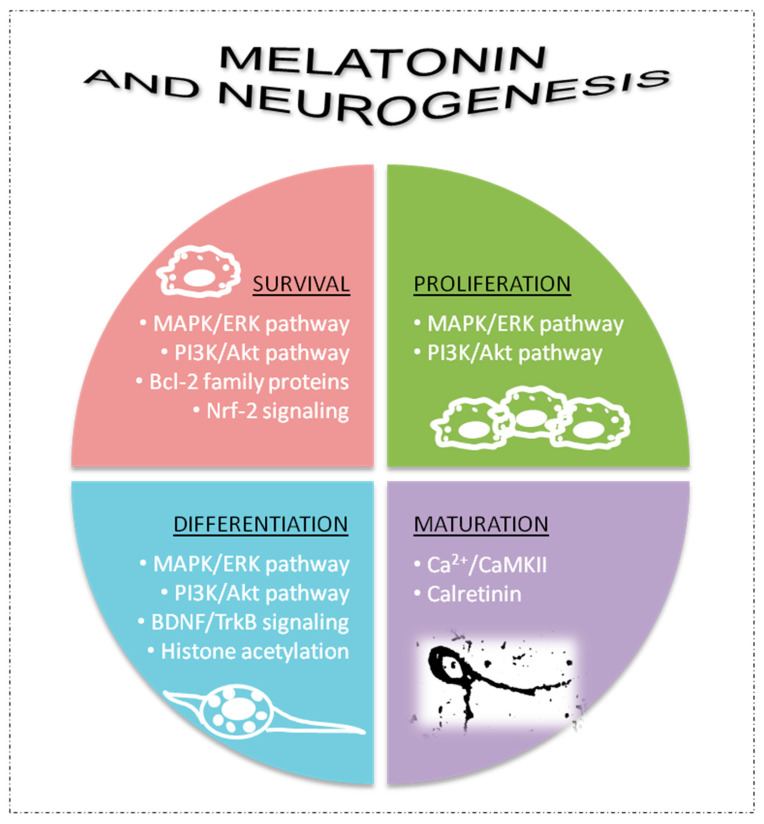
Some of mechanisms involved in the neurogenic potential of melatonin. AkT, protein kinase B; Bcl-2, B-cell lymphoma; BDNF, brain-derived neurotrophic factor; Ca^2+^/CaMKII, calcium/calmodulin-dependent kinase II; ERK, extracellular signal-regulated kinases; GDNF, glial cell line-derived neurotrophic factor; Nrf-2, nuclear factor erythroid 2-related factor 2; MAPK, mitogen-activated protein kinases; PI3K, phosphoinositide 3-kinase; TrkB, tropomycin-receptor kinase B.

**Figure 3 ijms-24-04803-f003:**
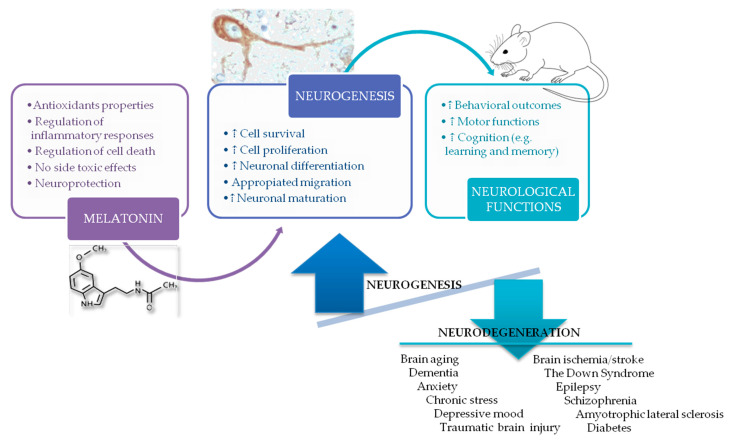
Melatonin may positively impact several neurological outcomes by correcting impairments that affect neurogenesis in the adult brain under different neuropathological conditions.

## Data Availability

Not applicable.

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
