# Peer review of "Benefits of the Neurogenic Potential of Melatonin for Treating Neurological and Neuropsychiatric Disorders"

_ijms, 2023, doi:10.3390/ijms24054803_

Round 1
Reviewer 1 Report
I have carefully read the manuscript “Benefits of the neurogenic potential of melatonin for treating neurological disorders” by Potes et al. The Authors describe the main findings on the neurogenic role of melatonin in different neurologic disease, such as Alzheimer´s disease, Down syndrome, brain injuries, epilepsy and others. In my opinion, the manuscript is well written and well structured.
In my opinion the manuscript might be useful to understand if melatonin can be used as therapeutic tool. Since the Authors declare “These beneficial effects of melatonin on different parameters of neurogenesis (survival, proliferation, differentiation, maturation, etc.) has (“have” is more appropriate) been demonstrated by using different concentrations of melatonin (from nM to µM) during acute treatments (from hours to days) in several models in vitro, including commercial cell lines (e.g. rat PC12 and mouse C17.2 cell lines), primary cultures of neural stem/progenitors cells obtained from brain or spinal cord of adult rats and mice, cultured mesenchymal stem cells from human amniotic fluid, induced pluripotent stem cells from mice as well as in adult 238 rat/mouse hippocampus organotypic cultures [55].” In my opinion, the role of melatonin on MSCs might be better described, also considering potential applications of MSCs in the regenerative medicine field. For this reason, I suggest a recent paper that may serve to explore signaling pathways involved in MSC neurogenic differentiation, on which melatonin might act.
Hardeland R. Melatonin and the Programming of Stem Cells. Int J Mol Sci. 2022 Feb 10;23(4):1971. doi: 10.3390/ijms23041971. PMID: 35216086; PMCID: PMC8879213.
At line 44, the term "this enzyme" is not appropriate.
Best Regards.
Reviewer 2 Report
This is a very interesting paper focusing on the benefits of melatonin in the treatment of brain disorders. The paper is well written and of interest for the journal. However, several minor changes are recommended before considering it for publication.
The main topic based on the title, seems to be the treatment of neurological disorders with melatonin. However, the authors report some evidence for the stress, depression and anxiety.
I recommend to rename the paper as "Benefits (...) for treating neuropsychiatric disorders. Perhaps, the authors can also divide the paper according to the main brain disorders they are focusing.
The authors conducted a narrative review. Although this is not a systematic review, I recommend to describe the methods: screening, selection processes, key words used to find articles.
Future perspectives and conclusions should be separated.
Melatonin has been also used to treat postmenopausal women with schizophrenia, which is a mental disorder considered also a Brain Disorder with hormonal effects. Few words focusing on the efficacy of melatonin could be useful. There are some reviews about the use and efficacy of melatonin in the treatment of schizophrenia.
Reviewer 3 Report
The review “Benefits of the neurogenic potential of melatonin for treating neurological disorders” by Potes et al. is well written and the review will be of interest to Biology readers. The authors did a comprehensive review of the potential benefit that melatonin may have in the treatment of certain neurological disorders.
Comment
Based on the pro-neurogenic properties of melatonin, the authors focus mainly on the effect of melatonin in the dentate gyrus neurogenesis and its beneficial administration in neurological disorders. However, the authors did not mention the impact that melatonin has on neurogenesis of the subventricular zone (SVZ), another neurogenic region that is essential for olfaction. Considering that several neurological disorders also affect olfaction, is it possible for authors to add some studies about the melatonin effect in SVZ neurogenesis and its benefit for neurological disorders?
